# The Role of Senescent Cells in Acquired Drug Resistance and Secondary Cancer in BRAFi-Treated Melanoma

**DOI:** 10.3390/cancers13092241

**Published:** 2021-05-07

**Authors:** Elizabeth L. Thompson, Jiayi J. Hu, Laura J. Niedernhofer

**Affiliations:** 1Department of Biochemistry, Molecular Biology, and Biophysics, University of Minnesota, Minneapolis, MN 55455, USA; hu000250@umn.edu (J.J.H.); lniedern@umn.edu (L.J.N.); 2Institute on the Biology of Aging and Metabolism, University of Minnesota, Minneapolis, MN 55455, USA

**Keywords:** melanoma, *BRAF* mutation, BRAF inhibitors, secondary cancer, resistance, senescence, senotherapeutics

## Abstract

**Simple Summary:**

Advances in melanoma treatment include v-Raf murine sarcoma viral oncogene homolog B (BRAF) inhibitors that target the predominant oncogenic mutation found in malignant melanoma. Despite initial success of the BRAF inhibitor (BRAFi) therapies, resistance and secondary cancer often occur. Mechanisms of resistance and secondary cancer rely on upregulation of pro-survival pathways that circumvent senescence. The repeated identification of a cellular senescent phenotype throughout melanoma progression demonstrates the contribution of senescent cells in resistance and secondary cancer development. Incorporating senotherapeutics in melanoma treatment may offer a novel approach for potentially improving clinical outcome.

**Abstract:**

*BRAF* is the most common gene mutated in malignant melanoma, and predominately it is a missense mutation of codon 600 in the kinase domain. This oncogenic *BRAF* missense mutation results in constitutive activation of the mitogen-activate protein kinase (MAPK) pro-survival pathway. Several BRAF inhibitors (BRAFi) have been developed to specifically inhibit *BRAF^V600^* mutations that improve melanoma survival, but resistance and secondary cancer often occur. Causal mechanisms of BRAFi-induced secondary cancer and resistance have been identified through upregulation of MAPK and alternate pro-survival pathways. In addition, overriding of cellular senescence is observed throughout the progression of disease from benign nevi to malignant melanoma. In this review, we discuss melanoma *BRAF* mutations, the genetic mechanism of BRAFi resistance, and the evidence supporting the role of senescent cells in melanoma disease progression, drug resistance and secondary cancer. We further highlight the potential benefit of targeting senescent cells with senotherapeutics as adjuvant therapy in combating melanoma.

## 1. Introduction

Melanoma is a cancer originating from melanocytes, the pigment producing cells in the skin [1]. In the United States alone, there will be an estimated 106,110 new cases and 7180 deaths from melanoma in 2021. Melanoma represents 5.6% of all new cancer cases in the US, and the rate of new cases has been increasing over the past 40 years. The risk of melanoma increases with age, but sex differences have been reported with women having a higher risk under age 50 and men having a higher risk over age 50 [2,3]. Other common risk factors for melanoma include ultraviolet (UV) light exposure, nevi (moles), fair skin, having prior melanoma or other skin cancer, family history of melanoma, or having a compromised immune system. Most nevi will never develop into melanoma; however, people with many nevi (greater than 50) or with congenital melanocytic nevi or dysplastic nevi are also at higher risk. In rare cases, increased risk of melanoma can be caused by an inherited DNA repair deficiency disorder called xeroderma pigmentosum [3,4].

Melanoma accounts for only 4% of all diagnosed skin cancers, but it is the most lethal type of skin cancer, resulting in 80% of skin cancer deaths [4]. The stage of melanoma diagnosis can determine both the survival rate and course of treatment [5]. Currently, the first treatment strategy for all stages of melanoma is surgical resection, which is highly curative for early stages of the disease. Melanoma is associated with a 5-year relative survival rate of 93.3%, but for distant metastatic disease the 5 year relative survival is only 29.8% [3]. If cancer cells have spread to the lymph nodes, then adjuvant or targeted therapy would be considered. New melanoma adjuvant treatments include immunotherapy with programmed cell death protein 1 (PD-1), programmed death ligand-1 (PD-L1), or cytotoxic T-lymphocyte associated protein 4 (CTLA4) inhibitors or targeted drug therapies with BRAF, MAPK/ERK kinase 1 (MEK), or stem cell factor receptor (c-KIT) inhibitors [6,7,8]. For advanced metastatic melanoma, other options include radiation and genotoxic chemotherapy [3], but they are typically less effective than the newer targeted treatments.

Despite melanoma having a high passenger mutation load due to ultraviolet (UV) mutagenesis, several cancer driving mutations have been identified. The three most prevalent genes mutated in melanoma are *BRAF*, neuroblastoma RAS viral oncogene homolog (*NRAS)*, and neurofibromatosis 1(*NF1)*, and all participate in the mitogen-activated protein kinase (MAPK) signaling cascade that regulates cell proliferation [9,10,11,12]. In fact, constitutive activating mutations in *BRAF* are the most common oncogenic mutations, present in 40–60% of all melanoma cases [11,13,14]. Additionally, *NRAS* mutations are found in 15–30% of melanoma patients [13,14,15], and *NF1* mutations are found in 12–18% of all melanomas [11,12]. Several other well know cancer genes that have been implicated in melanoma including phosphatase and tensin homolog *(PTEN)*, tumor protein p53 (*TP53)*, cyclin-dependent kinase inhibitor 2A (*CDKN2A)*, and mitogen-activated protein kinase kinase 1 (*MAP2K1)* [11]. Familial melanoma studies identified the *CDKN2A* locus, which encodes for protein p16^INK4a^ and p19^ARF^, and their loss of expression is common in melanoma [16,17]. Also, amplification of microphthalmia-associated transcription factor (*MITF*) occurs in 15% of melanomas, activating mutation of the receptor tyrosine kinase, KIT proto-oncogene (*KIT*) (~20–25% of melanomas), and germline melanocortin 1 receptor (MC1R) variants are lineage-specific casual alterations in melanoma [18,19,20]. Furthermore, novel driver mutations were identified in Rac family small GFPase 1 (*RAC1)*, AT-rich interactive domain-containing protein 2 (*ARID2)*, protein phosphatase 6 catalytic subunit (*PPP6C)*, and serine/threonine-protein kinase 19 (*STK19)* through genome-wide studies of a large number of melanomas [11,21]. Knowledge of the patient-specific mutations help guide treatment options with new targeted therapies.

Senescence has long been known as a double-edged sword in cancer biology. It is essential for cancer prevention and therapy effectiveness, but can also pave the way for resistance [22,23,24]. Senescence is a permanent cell cycle arrest induced by a variety of cell stressors including aging, genotoxic stress, or tissue injury. These cellular stressors lead to permanent activation of the DNA damage response (DDR) through ataxia telangiectasia mutated (ATM). The activated DDR results in the stabilization of tumor suppressor protein 53 (p53) and its transcriptional targets cyclin-dependent kinase inhibitor 1 (p21^CIP1^) and cyclin-dependent kinase inhibitor 2A (p16^INK4a^), resulting in cell cycle arrest through inhibition of cyclin dependent kinases which prevent the phosphorylation of retinoblastoma (RB) and entrance into S phase of the cell cycle [25,26,27,28]. Senescent cells are characterized by increased cell size, senescent-associated beta-galactosidase (SA-β-gal) activity [29], upregulation of anti-apoptotic pathways [30,31], decreased lamin B1 [32], and a senescence-associated secretory phenotype (SASP) [33]. The SASP is initiated by nuclear factor kappa B (NF-kB) signaling and is composed of proinflammatory cytokines, matrix metalloproteinases (MMPs) and growth factors [33,34]. These markers of senescence are routinely used to identify senescent cells; however, individually, none can confirm senescence.

In this review, we summarize the current understanding of *BRAF* mutations in melanoma. We explore the specific *BRAF^V600E^* mutation and discuss additional gene mutations often found in conjunction with *BRAF* mutations that lead to melanoma. We discuss current treatment with BRAF inhibitors (BRAFi) with secondary cancer side effects and resistance development. Finally, we evaluate the role of senescence in melanoma progression and treatment and propose intentionally targeting senescent cells as a novel treatment strategy to prevent secondary cancer and drug resistance.

## 2. BRAF Oncogene, Common Mutation BRAFV600E

BRAF is a cytoplasmic serine-threonine kinase in the MAPK pathway. Not only is *BRAF* the most common gene mutated in melanoma, but mutations in the kinase domain occur in over 60% of malignant melanomas. The predominate *BRAF* mutation site is valine 600, and 80–90% of the time it is the missense mutation V600E [35]. However, other missense mutations at this site are also found in melanoma including V600K (10–20%), V600R (5%), V600D (<5%) [14]. The amino acid change from valine to glutamic acid is a phosphomimetic mutation and results in constitute activation of BRAF and the MAPK pathway. This missense mutation is associated with younger age of onset of melanoma and more aggressive disease [14,36,37]. The *BRAF^V600E^* mutation is often found in benign and dysplastic nevi and these nevi typically remain in growth arrest for decades without advancing to malignancy [15,38]. These nevi express p16^INK4a^ and have increased senescence-associated acidic beta-galactosidase (SA-β-gal) activity, characteristic of senescent cells in permanent growth arrest. This senescence phenotype found in BRAF^V600E^ nevi is presumed to be caused by the oncogene-induced senescence (OIS) induced by a robust activation of the DNA damage response due to hyper-replication [39,40,41]. Therefore, due to OIS, the *BRAF^V600E^* mutation alone is not sufficient for malignancy and is often accompanied by additional driving mutation(s) [21,42,43]. Studies have found mutations in *PTEN*, a negative regulator of the phosphoinositide 3-kinase (PI3K) pathway, to have an occurrence as high as 40% with *BRAF* mutations [11]. Other common gene mutations implicated with *BRAF* mutations in melanoma include *CDKN2A*, tumor protein p53 *(TP53)* and telomerase reverse transcriptase (*TERT)* [11,44]. Melanoma has a high mutational burden compared to most other solid tumors and these mutations are primarily consistent with UV mutagenesis. The typical UV mutation signature is a cytidine to thymine transition occurring at dipyrimidine sites [11]. Interestingly, it is a thymine to adenine transversion mutation that results in the *BRAF^V600E^* mutation [45]. The *BRAF^V600E^* mutation site is directly adjacent to several dipyrimidines, and BRAF tandem mutations in melanoma are relatively common [46,47] (Figure 1). Thus, this cancer driving mutation could be caused by an error-prone polymerase producing a mutation near, but not at sites of UV-induced cyclobutene pyrimidine dimer DNA adducts [45,46,47,48]. The strong selective advantage of the mutation can then drive its common occurrence. Understanding that these *BRAF* mutations are likely driven by UV mutagenesis and often result in OIS is important for the prevention and treatment of melanoma.

Determining the presence of a *BRAF* mutation has become a standard of care for advanced melanoma and can help direct treatment with targeted therapies [7,14]. *BRAF* mutant melanoma is often more clinically aggressive and occurs in younger patients, therefore early identification is essential for optimal disease management [14]. Additionally, detection of *BRAF^V600E^* mutations in circulating tumor-derived DNA (ctDNA) in patient peripheral blood is correlated with tumor burden [49,50,51]. In some cases, increasing ctDNA *BRAF^V600E^* can be used as a biomarker of melanoma disease progression post therapy [49,50,51]. Furthermore, combining ctDNA *BRAF^V600E^* detection with other markers of disease progression such as lactate dehydrogenase (LDH) or matrix metalloproteinase-9 (MMP-9) can add prognostic value [49,50]. However, additional advancements in identifying early signs of disease progression are necessary for improved outcomes in *BRAF^V600E^* mutant melanoma.

The *Braf^V600E^* mutation was evaluated in mice and embryonic expression resulted in embryonic lethality. When the *Braf^V600E^* mutation was induced in melanocytes, highly pigmented lesions occurred within 3–4 weeks, however, these nevi were non-malignant with a senescent phenotype [52]. Similar to humans, the *Braf* oncogene induces melanocyte senescence in mice and can transform immortalized murine melanocytes [53,54]. Additionally, *Braf^V600E^* has been found with other driving mutations, such as *Pten*. Mice with *Pten* knockout alone do not develop melanoma, but when combined with *Braf^V600E^* (*Braf^V600E^*/*Pten*^−/−^ mice) there is rapid emergence of malignant lesions [52].

## 3. BRAF Inhibitors, Resistance and Secondary Cancer

Several BRAF inhibitors (BRAFi) that specifically target the *BRAF^V600^* mutant protein have been developed through structure-based drug design and including vemurafenib, dabrafenib and encorafenib. These drugs improve progression free survival (PFS) in Phase 3 randomized clinical trials in patients with metastatic melanoma and *BRAF^V600^* mutations. When compared to the standard of care chemotherapy, dacarbazine, the overall survival (OS) with vemurafenib was significantly increased from 9.7 months to 13.6 months and PFS increased from 1.6 months to 6.9 months [55]. Similarly, dabrafenib compared to dacarbazine treatment increased PFS in unresectable stage III melanoma from 2.7 to 5.1 months [56]. When the BRAFi, encorafenib was compared to vemurafenib with stage III and IV melanoma patients, the patients on either BRAF inhibitor had comparable PFS [57]. While initially very effective, melanoma resistance quickly develops to BRAFis, limiting their effectiveness.

Genetic alterations providing resistance to BRAFi are found in the majority of resistant tumors (Figure 2) [58,59,60,61,62,63,64]. The most common genetic changes for resistance result in reactivation of the MAPK pathway. These occur through *BRAF* amplification, *BRAF* splice variants, or additional mutations in *BRAF*, *RAS* or *MEK* [58,59,62]. Loss of *CDKN2A* was also commonly found in BRAFi resistant melanomas and is linked to the MAPK pathway as an inhibitor of the downstream effectors cyclin D and cyclin-dependent kinase 4 (CDK4) [59,64]. The phosphoinositide 3-kinase/protein kinase B/mammalian target of rapamycin (PI3K/ATK/mTOR) pathway was implicated as the second major pathway upregulated in BRAFi resistance. Several mutations in this pathway including loss of *PTEN* or upregulation of *AKT* or receptor tyrosine kinases (RTKs) are commonly found in resistant melanoma [43,52,58,62,65,66,67]. Overall, most BRAFi resistant melanomas have reactivated the MAPK pathway or upregulated the PI3K/AKT/mTOR pathway through specific mutations. However, altered epigenetic regulation and evasion of the immune system also occur to promote tumor proliferation and survival.

Several non-genetic mechanisms also contribute to melanoma progression and resistance including phenotypic switching, tumor microenvironment, inflammation, and epigenetic changes (Figure 3A). Rambow et al. investigated phenotypic switching associated with minimal residual disease (MRD) and drug resistance in melanoma [68]. The authors used patient-derived xenographs from *BRAF^V600^* mutated tumors and subsequently treated the mice with BRAFi/MEKi. They found 60% of resistant cells had mutations in MAPK or PI3K/AKT pathways that helped drive drug resistance. However, these mutations were not observed at the MRD phase, indicating drug tolerance is also driven by epigenetic and environmental cues. Retinoid X receptors (RXR) signaling was identified as driving treatment resistance, in particular RXRγ [68]. While knockdown of RXR receptors is associated with senescence, primarily through RARα and RARβ, the role of RXRγ in senescence is not clearly defined [69].

Microphthalmia-associated transcription factor (MITF) has also been used in melanoma as an indicator of melanoma cell phenotype switching. The emergence of a drug resistant and invasive phenotype is associated with low expression of MITF, and in general, MITF is downregulated as disease progresses [68,70,71]. However, some heterogeneity in MITF expression is observed in MRD [68]. Decreased expression of MITF is also associated with senescence and increased genomic instability and the emergence of aggressive metastatic cells [72,73]. Proinflammatory cytokines released after treatment such as transforming growth factor beta (TGFβ), platelet-derived growth factor (PDGF), tumor necrosis factor alpha (TNFα), interleukin-1b (IL-1β), and interleukin-6 (IL-6) that can activate myofibroblast and contribute to chronic inflammation, fibrosis, and a pro-tumor microenvironment [71,74]. Hepatocyte growth factor secretion by cancer associated fibroblast (CAFs) can also activate receptor tyrosine kinases (RTKs) and the MAPK and PI3K/AKT pathways driving drug resistance [75]. All of these cytokines associated with creating the proinflammatory and pro-tumor environment are also SASP factors that could be secreted by senescent cells in the tumor microenvironment [76,77,78].

High mobility group box protein 1 (HMGB1) has also been implicated in melanoma progression with higher expression correlating with poor survival [79]. However, excretion of HMGB1 is also associated with a proinflammatory and anti-tumor response through activation of dendritic cells and induction of T-cell activation after BRAFi and MEKi treatment [80]. HMGB1 is a damage-associated molecular pattern (DAMP) that is also secreted by senescent fibroblasts induced by several different types of cellular stress including OIS, protease inhibitor, and ionizing radiation (IR) [77].

Immune escape is a common occurrence for patients treated with BRAFi. Immune escape and metastatic disease are associated with increased CEACAM1, an intercellular adhesion protein that has immune modulation functions [81,82,83]. Furthermore, CEACAM1 expression is required for senescence maintenance [84] and detection of high levels of CEACAM1 in melanoma is associated with oxidative stress, immune dysfunction and metastatic disease [81,82,83,85]. BRAFi treatment initially downregulates CEACAM1, but its expression is restored with the development of drug resistance [85], indicating a contribution of senescent cells to drug resistance.

Finally, epigenetic factors contribute to melanoma progression and drug resistance. One example is enhancer of zeste homologue 2 (EZH2), which is a histone methyltransferase that can trimethylate lysine 27 in histone 3 (H3K21me3) and repress transcription. In melanoma, EZH2 expression is associated with poor prognosis [86]. Additionally, depletion of EZH2 can activate p21^CIP1^ and result in senescence [87]. Furthermore, alterations in expression of histone deacetylases (HDACs) and histone acetyltransferases can result in epigenetic changes leading to BRAFi resistance. Some resistant melanomas were found to downregulate HDACs (Sirtuin 2 and Sirtuin 6) and histone acetyltransferase (HAT1), or to upregulate HDAC8 [58]. Similarly, components of the SASP are altered with HDAC inhibition to senescent fibroblasts, promoting tumor growth [88]. Collectively, these studies establish parallels between the non-genetic mechanisms of resistance in melanoma and some of the pro-tumor characteristics of senescence cells (Figure 3A,B).

One of the most concerning side effects of BRAFi is non-melanoma skin lesions, including cutaneous squamous cell carcinoma (cSCC), which occur in 15–20% of patients. Although cSCC is another form of skin cancer, it arises from keratinocytes instead of melanocytes. Importantly, BRAFi treatment of cells with wild type BRAF leads unexpectedly to activation of the MAPK pathway, and this upregulation is fundamental for cSCC development [89,90,91]. This paradoxical activation of the MAPK pathway with BRAFi is often found in keratinocytes with existing *RAS* mutations. In fact, RAS mutations are found in up to 60% of BRAFi therapy-induced secondary cSCC [61,91,92]. Oncogenic RAS with BRAFi has been found to activate the MAPK pathway by signally through CRAF, an analog of BRAF [61,90,91]. While the BRAFi have very low target inhibition of wild type BRAF, this activation of CRAF can result in increased proliferation and tumorigenesis in *RAS* mutant cells (Figure 2) [90,93]. Additionally, there have been reports of melanoma patients treated with BRAFi that have developed non-cutaneous RAS driven cancers including leukemia and colon cancer [94,95,96]. Another potential driving mechanism for secondary cSCC is human papilloma virus (HPV) infection [97,98]. Multiple variations of beta-HPV have been found in cSCC that were induced by BRAFi treatment [99]. The concern over paradoxical activation of the MAPK pathway, secondary cancer development and resistance with BRAFi led to combination therapy testing.

## 4. Addressing Secondary Cancer and Resistance to BRAF Inhibitors

The use of MEK inhibitors (MEKi) in combination with BRAFi was to prevent the paradoxical and reactivation of the MAPK pathway that leads to resistance and cSCC development [58,93,100,101]. The addition of MEKi were successful in reducing the incidence of cSCC and increasing PFS, but resistance routinely still develops. The combination treatments are Vemurafenib with Cobimetinib [102], Dabrafenib with Trametinib [103], and Encorafenib with Binimetinib [57]. While the MEK inhibitors reduce secondary cSCC, the two drugs target the same molecular pathway, increasing the chances of resistance and secondary cancer to occur through alternative pathways (Figure 2).

## 5. Senescence in Tumorigenesis and Melanoma

Senescence is typically seen as a repressor of tumorigenesis by halting growth in premalignant cells [104]. However, when senescence cells are not cleared by immunosurveillance, a pro-inflammatory environment can emerge that drives aging pathologies, including cancer [22,104]. For example, while transient senescence is associated with efficient wound healing, persistent senescent cells are associated with chronic wounds [105,106]. With increased age, there is an increase in senescent cells and an increase risk of cancer. The correlation of cancer with senescent cell burden was demonstrated using a genetic mouse model for inducible elimination of p16^Ink4a^-positive senescent cells. The continuous elimination of senescent cells in adult mice delayed the onset of age-associated cancer [107]. Furthermore, there is even evidence indicating the necessity of overriding senescence as a requirement for malignancy and increased cancer aggressiveness [60,108,109,110,111].

There are several characteristics of senescent cells that can contribute to tumorigenesis (Figure 3B). First, senescent cells are in a state of chronic DNA damage response (DDR) with an increase in DNA damage foci [112,113]. This chronic DDR leads to mitochondrial dysfunction and increased reactive oxygen species (ROS) that create a feedback loop to help maintain the DDR [114,115]. Persistent DNA damage and ROS production can precipitate intracellular changes that promote malignant transformation. Second, the senescence-associated secretory phenotype (SASP) is a mixture of cytokines, chemokines, growth factors, and proteases secreted from senescent cells and is credited with creating a pro-inflammatory and pro-tumor microenvironment [33]. The SASP has both paracrine and autocrine effects in maintaining senescence and spreading senescence to neighboring cells. Additionally, the SASP can induce both epithelial to mesenchymal transition (EMT) and reprogramming of cancer cells to a more stem-like phenotype [33,116,117]. Immunosenescence refers to age-related dysfunction of the immune system caused at least in part by immune cell senescence and low-grade chronic inflammation due to SASP (inflammaging). The SASP of the tumor microenvironment can drive both inflammaging and immunosenescence, resulting in reduced clearance and accumulation of senescent cells and tumor escape from immunosurveillance [74,118,119,120,121,122]. This impairment of the immune system also results in compromised immune surveillance that is vital for tumor progression and metastasis.

There is evidence of senescence or a senescence-like phenotype throughout the progression of BRAF mutant melanoma. A senescence phenotype is observed in pre-neoplastic cells where benign nevi express characteristic senescence markers with increased SA-β-gal activity, increased expression of p16^INK4a^, decreased lamin B1, and these markers decrease as the tumor progresses [39,123,124]. Additionally, BRAF^V600E^ melanoma cells have increased senescence characteristics with increased SA-β-gal activity and expression of SASP factors, including IL-8 and TGFβ, compared to wild type BRAF melanoma cells. Moreover, therapy induced senescence (TIS) has been observed in vitro with Vemurafenib treatment resulting in increased SA-β-gal activity in both sensitive and resistance BRAF^V600E^ mutant melanoma cells [60]. Of further interest, the secondary cSCC from BRAFi-treated melanoma patients stains strongly for p16^INK4a^ [125,126]. Additionally, a senescent phenotype is often observed in HPV infected cells, HPV positive tumors, and from RAS oncogene expression [97,98,99,124,127], which are commonly found in BRAFi-induced secondary cSCC. Furthermore, BRAFi treatment initially increases the antitumor immunogenicity of melanoma, but when resistance develops the tumors revert to a low immunogenic state. This lower immunogenic state of the BRAF-resistant cells has fewer tumor infiltrating T-cells and NK cells and they are less effective at recognizing and killing the cancer cells [58,128], which are all characteristics of immunosenescence. Taken together, these studies demonstrate the potential role of senescence within melanocytes and the skin microenvironment that are influencing melanoma progression and secondary cancer occurrence.

## 6. Future Therapy Directions

While targeted therapies like BRAFi are highly effective therapeutics at first, melanoma resistance routinely develops. Cross-talk between pro-survival pathways is well-established and upregulation of alternate pathways, primarily PI3K/AKT/mTOR pathway, is consistently seen in resistance to BRAFi [129]. In addition, branched evolution of resistance with multiple pathways of acquired resistance have been found within the same melanoma patient [59]. Furthermore, BRAFi resistant BRAF^V600E^ melanoma cells can even become dependent on the BRAF inhibitor for proliferation [60]. All these examples highlight the necessity for adjuvant treatments that can target several anti-apoptotic pathways to prevent drug resistance, such as senotherapeutics.

Senotherapeutics are a class of drugs that targets senescent cells and include senolytics and senomorphics. Senolytics selectively kill senescent cells and senomorphics suppress the SASP without inducing apoptosis. However, the distinction between these two drug classifications can sometimes be cell-type specific with the same drug capable of having senolytic function on one cell type and senomorphic on another [130]. Senolytics have been shown to combat aging and improve healthspan by reducing frailty and attenuating common age-associated morbidities [130,131,132,133]. Now senolytics are being considered and used in combination therapy for cancers [119,134,135,136]. Senotherapeutics could aid in the elimination of senescent cells after therapy-induced senescence (TIS) to reduce the emergence of drug resistance and cancer recurrence [134,135]. Several senotherapeutics have the added benefit of targeting multiple anti-apoptotic pathways that are employed by senescent cells to stay alive [131]. In addition, several senotherapeutics are flavonoids found in many fruits and vegetables and, therefore, expected to have lower toxicity than chemotherapy drugs that target proliferating cells. In theory, senotherapy could aid the immune system in clearing BRAFi-induced senescent cancer cells, reducing the possibility of acquiring additional adaptations for resistance and secondary cancers.

Several senotherapeutics have been tested with melanoma cells in vitro with encouraging results and some clinical trials with melanoma patients have been conducted (Table 1). One senolytic being researched with melanoma is dasatinib, a broad-specificity receptor tyrosine kinase (RTK) inhibitor. EGFR is a RTK that is commonly upregulated as a mechanism of resistance to BRAFi [66], therefore, dasatinib has both senolytic activity [133] and targets RTKs in BRAFi resistance. Importantly, dasatinib inhibits the proliferation and invasion of even BRAFi-resistant melanoma cells [67]. Another senolytic, fisetin is a flavonoid found to extend the health and lifespan of both progeroid and aged wild-type mice [132]. Fisetin has been tested in melanoma cells in an in vitro 3-D model system and found to promote tumor regression [137]. Fisetin has inhibitory effects on the PI3K/AKT/mTOR pathway and demonstrated efficiency at inhibiting multiple cancers and specifically melanoma [138,139]. Quercetin is another flavonoid and senolytic that has been found to work synergistically with dasatinib to combat age-associated frailty and extend healthspan in pre-clinical models [130]. Quercetin inhibits the growth, invasiveness, and metastatic potential of melanoma cell lines [140]. Additionally, quercetin can be metabolized by tyrosinase, which is expressed in melanocytes, into additional anti-cancer compounds, potentially increasing its potency specifically for melanoma [141,142]. Finally, piperlongumine, a natural extract from the Piper Longum pepper plant induces apoptosis of melanoma cells [143]. The benefits of the highlighted senolytic drugs in melanoma treatment is two-fold. First, senolytics could aid clearance of senescent tumor cells to prevent resistant cells from emerging. Second, they target senescent cell anti-apoptotic pathways that are involved in secondary cancer and development of drug resistance [131]. Intermittent dosing of BRAFi has demonstrated improved efficacy against some therapy resistant melanoma [144], and these resistant tumors could potential benefit from alternating BRAFi treatment with a senotherapy recently termed the “one-two punch” [145]. This “one-two punch” treatment strategy suggests treating with a tumor targeted therapy to induce senescence and following with a senolytic to help clear the senescent cells [145]. Some of the long-term effects of chemotherapy treatment on cancer survivors include increased incident of age-associated diseases linked to senescence such as cardiovascular disease, neurodegeneration, and secondary cancer. Thus, the use of senotherapeutics as part of cancer management could improve cancer treatment effectiveness and the healthspan of cancer survivors.

## 7. Conclusions

Early-stage melanoma is curable, but once metastases have occurred, the chance of survival diminishes significantly. Several targeted therapies have been developed toward oncogenic BRAF in malignant melanoma that significantly improve PFS, but resistance often develops. Throughout the progression of disease and treatment of BRAF^V600E^ driven melanoma, senescence induction and escape are recurring themes from cancer initiation to secondary skin cancers (cSCC), and drug resistance (Figure 4). The cancer cells must escape OIS induced by the BRAF^V600E^ mutation in stable nevi to become malignant. The BRAFi therapy-induced senescence (TIS) in the cancer cells, and the development of secondary cSCCs with BRAF inhibitor treatment relies on upregulation of the MAPK pathway or alternative pathways in keratinocytes that often have senescence driving alterations such has RAS oncogene activation or HPV infection. Finally, the emergence of resistance occurs through upregulation of anti-apoptotic pathways like the MAPK pathway and AKT/PI3K/mTOR to override senescence induction. This repeated failure of senescence maintenance to control disease progression represents an opportunity for combining the use of senotherapeutics in melanoma treatment regimens for BRAF^V600E^ mutated cancer. Particularly considering the evidence of cancer cells that escape from senescence can be primed as more stem-like, aggressive, and drug resistant. Several senotherapeutics have been tested on melanoma cells and show promising results, but further in-vivo studies are necessary to confirm these results and advance senotherapeutics for melanoma treatment.

## Figures and Tables

**Figure 1 cancers-13-02241-f001:**
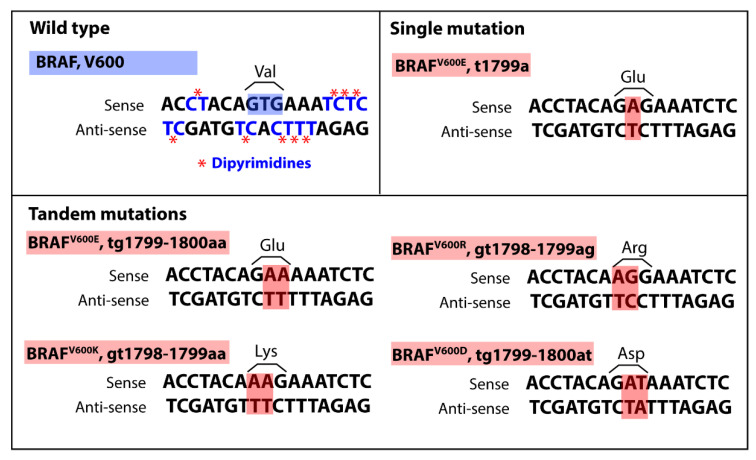
**Common BRAF codon 600 missense mutations found in melanocytic lesions. Wild type***BRAF* encoding valine at codon 600 with dipyrimidines highlighted in blue with *. Dipyrimidines are common sites of UV mutagenesis resulting in C-T transitions and are prevalent surrounding BRAF codon 600. The **Single mutation**
*BRAF^V600E^* T-A transversion mutation is the most common mutation in melanoma and is not typical of UV mutagenesis. **Tandem mutations** are commonly found in BRAF at codon 600 and include *BRAF^V600E^*, *BRAF^V600R^*, *BRAF^V600K^*, and *BRAF^V600D^*. All the Tandem mutations except BRAF^V600D^ contain one C-T transition at a dipyrimidine site within the tandem mutation. The surrounding dipyrimidine sites and common tandem mutations highlight the possibility of UV mutagenesis and error-prone polymerase incorporation of incorrect bases at *BRAF* codon 600. BRAF, v-Raf murine sarcoma viral oncogene homolog B.

**Figure 2 cancers-13-02241-f002:**
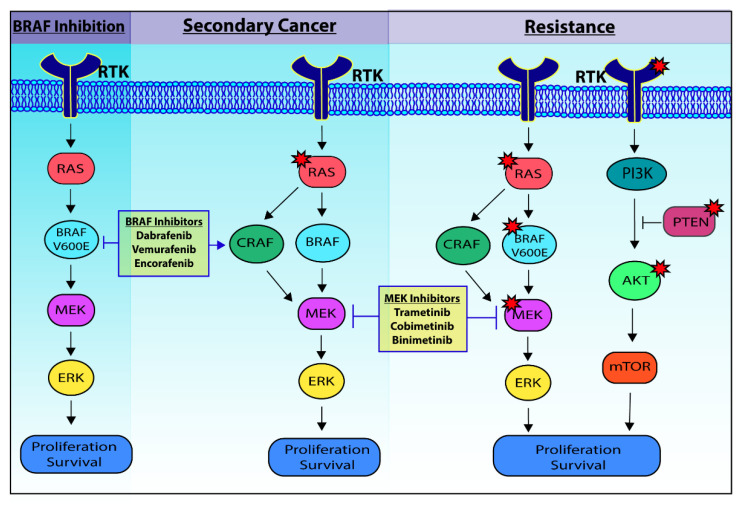
**BRAF inhibition and genetic mechanisms of secondary cancer and resistance. BRAF Inhibition:** BRAF^V600E^ results in constitutive activation of the pro-survival MAPK pathway. BRAF inhibitors target the oncogenic *BRAF* mutation and initially are highly effective. **Secondary Cancer**: BRAF inhibitors can paradoxically activate the MAPK pathway in cells with wild type BRAF and oncogenic *RAS* mutations by signaling through CRAF, resulting in cSCC development. The occurrence of cSCC is reduced when MEK inhibitors are used to further restrict the activation of the MAPK pathway. **Resistance** to both BRAF inhibition and MEK inhibition predominately develop through *RAS*, *BRAF^V600E^*, *MEK*, RTK, PTEN, or *AKT* mutations indicated by red stars. These mutations either reactivate the MAPK pathway or upregulate the PI3K/AKT/mTOR pro-survival pathway. cSCC, cutaneous squamous cell carcinoma; RTK, receptor tyrosine kinase; MAPK, mitogen activated protein kinase; BRAF, v-Raf murine sarcoma viral oncogene homolog B; ERK, extracellular signal-regulated kinase; MEK, MAPK/ERK Kinase 1; RAS, rat sarcoma; AKT, protein kinase B; PI3K, phosphoinositide 3-kinase; mTOR, mammalian target of rapamycin; PTEN, phosphatase and tensin homolog.

**Figure 3 cancers-13-02241-f003:**
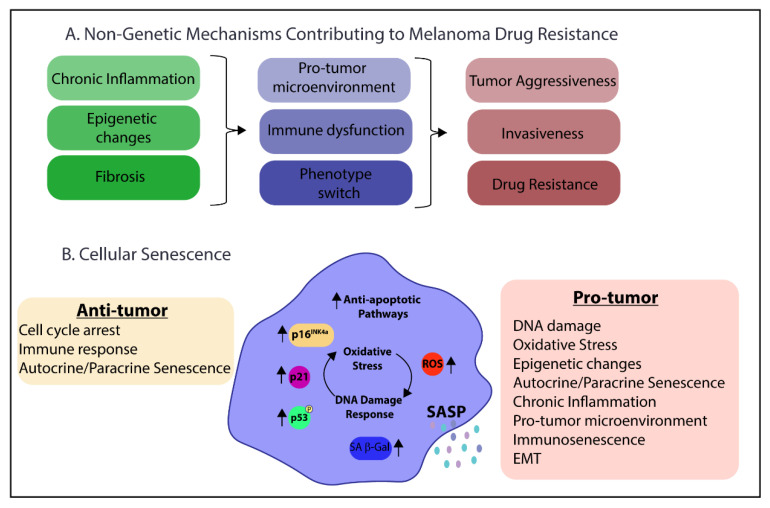
Parallels between (**A**) Non-genetic mechanisms contributing to melanoma drug resistance and (**B**) Cellular senescence pro-tumor characteristics. p16^INK4a^: cyclin-dependent kinase inhibitor 2A; p21: cyclin-dependent kinase inhibitor 1A; p53: tumor protein p53; ROS: reactive oxygen species, SA- β-gal: senescence-associated beta-galactosidase; SASP: senescence associated secretory phenotype; EMT: epithelial to mesenchymal transition.

**Figure 4 cancers-13-02241-f004:**
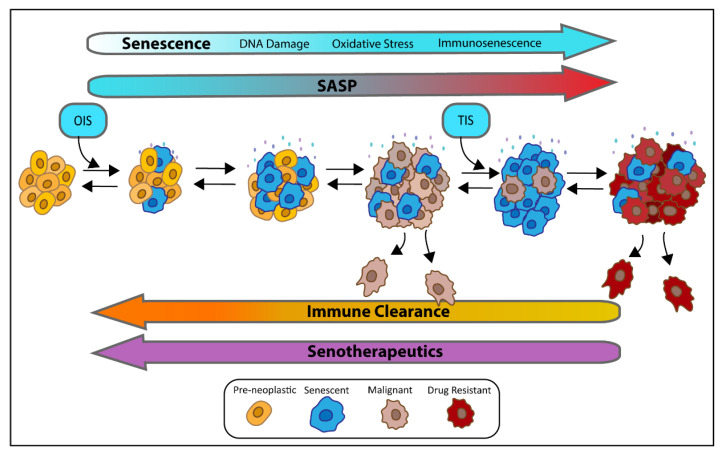
**The role of senescent cells in cancer progression and resistance.** Pre-neoplastic lesions with BRAF^V600E^ or RAS mutations can acquire senescent cells through oncogenic induced senescence (OSI). The OIS cells can spread senescence to nearby cells through their senescent associated secretory phenotype (SASP). If these senescent cells are not cleared by the immune system, the SASP can produce a pro-inflammatory and pro-tumor microenvironment with increased DNA damage, oxidative stress, and immunosenescence resulting in malignant transformation. Therapy induced senescence (TIS) by BRAFi is initially beneficial in combating malignant melanoma, but can lead to increased senescence, drug resistance, and aggressive, stem-like tumor phenotype. Senotherapeutics can reverse all phases of disease progression by targeting pro-survival pathways, reducing SASPs and/or by aiding the immune system in clearing senescent cells.

**Table 1 cancers-13-02241-t001:** Clinical trials and other studies on applying senotherapeutics to melanoma. HSP: heat shock protein; Bcl-2/Bcl-xL: B-cell lymphoma 2/B-cell lymphoma extra-large; HDAC: histone deacetylase; OXR1: oxidation resistance 1; BET: bromodomain and extraterminal domain; PI3K: phosphoinositide 3-kinase; AKT: protein kinase B; mTOR: mammalian target of rapamycin; MIC: melanoma initiating cells This table is not an all-inclusive list of studies with senotherapeutics for melanoma, and a more extensive list of senotherapeutics used in additional cancers can be found in Prasanna, et al. [145].

Drug	Mechanisms of Action	Treatment	Developmental Stage
Alvespimycin	HSP inhibitor	Alvespimycin hydrochloride	**Clinical Trail:** Phase 1; NCT00089362; Metastatic or unresectable solid tumors including melanoma
		Alvespimycin hydrochloride	Phase 1; NCT00248521; Adult solid tumor including melanoma [146]
		Alvespimycin hydrochloride	**Discovery Phase:** human melanoma cell line [147]
Tanespimycin	HSP inhibitor	Tanespimycin	**Clinical Trail:** Phase 2; NCT00087386; Recurrent or phase III, IV melanoma
		Tanespimycin	Phase 1; NCT00004065; Refractory advanced solid tumors including melanoma or hematologic cancer
		Tanespimycin	Phase 2; NCT00104897; Metastatic malignant melanoma [148]
		Tanespimycin	Phase 2; Metastatic Melanoma [149]
		Tanespimycin and Sorafenib	Phase 1; Melanoma, renal cancer and colorectal cancer [150]
Digoxin	Na+/K+ ATPase inhibitor	Trametinib and Digoxin	**Clinical Trail:** Phase 1; NCT02138292; Unresectable or metastatic BRAF wild-type melanoma [151]
		Vemurafenib and Digoxin	Phase 1; NCT01765569; Advanced *BRAF^V600^* mutant melanoma
Navitoclax (ABT-263)	Bcl-2/Bcl-xL inhibitor	Dabrafenib, trametinib, and navitoclax	**Clinical Trail:** Phase 1/2; NCT01989585; BRAF mutant melanoma or unresectable or metastatic solid tumors
		Novitoclax and selumetinib	**Discovery Phase:** Melanoma cell lines [152]
Dasatinib	Pan receptor tyrosine kinase inhibitor	Dasatinib	**Clinical Trail:** Phase 2; NCT00700882; Melanoma (Skin) [153]
		Dendritic cell Vaccines + Dasatinib	Phase 2; NCT01876212; Metastatic melanoma
		Dasatinib and Dacarbazine	Phase 1/2; NCT00597038; Metastatic Melanoma
		Dasatinib	Phase 2; NCT00436605; Unresectable stage III melanoma or stage IV melanoma
		Dasatinib	Phase 1; Advanced melanoma [154]
		Dasatinib and Dacarbazine	Phase 1; Metastatic melanoma [155]
		Dasatinib	**Discovery Phase:** Melanoma cell lines [67]
Panobinostat (LBH589)	Pan HDAC inhibitor	Panobinostat	**Clinical Trail:** Phase 1; NCT01065467; Metastatic Melanoma
		Panobinostat and Ipilimumab	Phase 1; NCT02032810; Unresectable stage III/IV melanoma
		Temozolomide, Decitabine, Panobinostat	Phase 1/2; NCT00925132; Metastatic Melanoma [156]
		Panobinostat (LBH589)	Phase 1; metastatic melanoma [157]
**Curcumin analog, EF24**	Promote degradation of anti-apoptotic Bcl-2 proteins	EF24	**Discovery Phase:** Malignant melanoma cell lines [158]
**Piperlongumine**	Targets OXR1	Piperlongumine	**Discovery Phase:** Human melanoma cell [143]
**Ouabain**	Na+/K+ ATPase inhibitor	Ouabain	**Discovery Phase:** Malignant melanoma cell lines [159,160]
**ABT-737**	Bcl-2/Bcl-xL inhibitor	ABT-737 and PLX4720	**Discovery Phase:** Human melanoma cell lines and primary melanoma cell culture [161]
		ABT-737 and GSI (γ-Secretase Inhibitor)	Non-MIC (bulk of melanoma) and MICs [162]
**JQ1**	BET inhibitor	JQ1 and vemurafenib	**Discovery Phase:** BRAF mutant vemurafenib-resistant melanoma cells [163]
**Quercetin**	Activates estrogen receptors and inhibits PI3 kinase	Quercetin	**Discovery Phase:** Melanoma cell lines [140,141,142,164,165]
**Fisetin**	Blocks PI3K/AKT/mTOR pathway	Fisetin	**Discovery Phase:** Melanoma cell lines [137,138,166,167,168]

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
