# Peer review of "The Role of Senescent Cells in Acquired Drug Resistance and Secondary Cancer in BRAFi-Treated Melanoma"

_cancers, 2021, doi:10.3390/cancers13092241_

Round 1

Reviewer 1 Report

The review of Thompson et al focuses on the role of senescence in response of melanoma to BRAFi. Overall, the review must present more current knowledge on the mechanisms of drug resistance in melanoma, and extend the main body of the text on available papers in the scope of this manuscript.

Specific comments:

  1. line 17 (abstract) is not clear - amino acid cannot be mutated (it is oversimplification)
  2. In the introduction, the Authors focuses only on the genetic alterations accompanying the acquisition of resistance. This is not up-to-date knowledge! There are a number of papers of substantial phenotypic plasticity that underlies drug resistance in melanoma.
  3. Fig. 1 is an overpresentation of simple and well-known data.
  4. A table with the characteristics of senolytic drugs applicable to melanoma would be beneficial.
  5. The Authors could discuss the induction of senescence in response to BRAFi/MEKi in the short run. There are more papers on this issue, including those validating the contribution of certain genetic variants with regard to responsiveness to BRAFi/MEKi with senescence e.g, EZH2.

Author Response

Thank you for your feedback.  We have responded to you comments below.

The review of Thompson et al focuses on the role of senescence in response of melanoma to BRAFi. Overall, the review must present more current knowledge on the mechanisms of drug resistance in melanoma, and extend the main body of the text on available papers in the scope of this manuscript.

Specific comments:

1. line 17 (abstract) is not clear - amino acid cannot be mutated (it is oversimplification)

Response: This has been corrected to refence mutation of the gene not the amino acid (line 17).

2. In the introduction, the Authors focuses only on the genetic alterations accompanying the acquisition of resistance. This is not up-to-date knowledge! There are a number of papers of substantial phenotypic plasticity that underlies drug resistance in melanoma.

Response: We expanded the information on non-genetic contributions to drug resistance (lines 206-259) and added a new figure (Figure 3) to directly address the lack of non-genetic mechanisms of resistance.

3. Fig. 1 is an overpresentation of simple and well-known data.

Response: We can move this figure to the supplement, however this figure touches on two major contributors to senescence: genome instability and oncogenic induced replication stress. It also explains how these two drivers of senescence occur in UV damaged skin and in melanocytes and why BRAFV600 missense substitutions are commonly found. A sentence clarifying this point was added (line 135-137).

4. A table with the characteristics of senolytic drugs applicable to melanoma would be beneficial.

Response: We created Table 1 of clinical trials and pre-clinical work with senotherapeutics for melanoma specifically.  We also reference a recent publication with a table summarizing cancer related clinical trials with senotherapeutics (line 432).

5. The Authors could discuss the induction of senescence in response to BRAFi/MEKi in the short run. There are more papers on this issue, including those validating the contribution of certain genetic variants with regard to responsiveness to BRAFi/MEKi with senescence e.g, EZH2.

Response: We included some references on EZH2 expression and melanoma progression, but not with genetic variants, as epigenetic modifications leading to disease resistance (lines 248-252).

Reviewer 2 Report

In this review, the authors discuss melanoma BRAF mutations, the genetic mechanism of BRAFi resistance, and the evidence supporting the role of senescent cells in melanoma disease progression, drug resistance and secondary cancer. The authors further highlight the potential benefit of targeting senescent cells with senotherapeutics as adjuvant therapy in combating melanoma.

The review has addressed the latest advancement in the field, however, given the authors do suggest the emergence of resistance occurs through upregulation of anti-apoptotic pathways like the MAPK pathway and AKT/PI3K/mTOR to override senescence induction. This repeated failure of senescence maintenance to control disease progression represents an opportunity for combining the use of senotherapeutics in melanoma treatment regimens for BRAFV600E mutated cancer. Particularly in light of the evidence of cancer cells that escape from senescence can be primed as more stem-like, aggressive, and drug resistant. Given several senotherapies has already been tested in melanoma as well as other cancer subtypes, it will be worthwhile mentioning about clinical trials that are in progress or approved drugs. The mention will enhance the readers and update the advancements.

Author Response

Thank you for your feedback.  We have responded to your comments below.

The review has addressed the latest advancement in the field, however, given the authors do suggest the emergence of resistance occurs through upregulation of anti-apoptotic pathways like the MAPK pathway and AKT/PI3K/mTOR to override senescence induction. This repeated failure of senescence maintenance to control disease progression represents an opportunity for combining the use of senotherapeutics in melanoma treatment regimens for BRAFV600E mutated cancer. Particularly in light of the evidence of cancer cells that escape from senescence can be primed as more stem-like, aggressive, and drug resistant. Given several senotherapies has already been tested in melanoma as well as other cancer subtypes, it will be worthwhile mentioning about clinical trials that are in progress or approved drugs. The mention will enhance the readers and update the advancements.

Response: We created Table 1 of clinical trials and pre-clinical work with senotherapeutics for melanoma specifically.  We also reference a recent publication with a table summarizing cancer related clinical trials with senotherapeutics (line 432).

Reviewer 3 Report

Thompson et al. “The role of senescent cells in acquired drug resistance and secondary cancer in BRAFi-treated melanoma”

Overall, this is a well written article that places emphasis on RAFi induced senescent cells as a crucial hurdle for long-term clinical benefit.  While this is an important topic, some concerns are outlined below:

  • There seemed to be quite a bit of broad melanoma background/information and relatively little specifically discussing targeting senescence itself. The interesting sensotherapeutic commentary started in earnest on line 286 for 1 paragraph. 
  • I struggle with Figure 1’s contribution to this article
  • While I understand there are distinct molecular differences, there was no discussion of the relationship between senescence and quiescence. There are a number of publications outlining how targeted inhibitors leave behind a cancer stem cell enriched/quiescent-like “minimal residual disease” (Rambow et al. Cell 2018) - these underlying concepts work hand-in-hand.
  • How does intermittent dosing (Das Thakar et al Nature 2013 and others) play a role in senescence?
  • On line 179, the authors mention that MEKi was combined with RAFi to reduce paradoxical activation and secondary cancers. This needs a citation, and is not the sole reason for combination treatment.  Also, there is no mention of “paradox breaking” RAFi, or pan-RAFi.
  • While touched on at the end of the “Senescence in tumorignesis and melanoma” section, the role of the immune system should be expanded. How might RAFi induced immunogeneic cell death play a role in senescence (Erkes et al. Cancer Discovery 2020)? How do checkpoint inhibitors contribute to or reduce the number/viability of senescent cells?

Minor points:

  • Line 49-50: Survival of less than 10% seems to be low.
  • Line 121-123: This part is not clear – what is embryonic lethal?
  • Line 252: Define immunosenescence.

Author Response

Thank you for your feedback.  We have responded to your comments below.

Overall, this is a well written article that places emphasis on RAFi induced senescent cells as a crucial hurdle for long-term clinical benefit.  While this is an important topic, some concerns are outlined below:

  • There seemed to be quite a bit of broad melanoma background/information and relatively little specifically discussing targeting senescence itself. The interesting sensotherapeutic commentary started in earnest on line 286 for 1 paragraph. 

Response: We added a paragraph introducing senescence to the introduction section (lines 81-96), and the senescence theme is incorporated throughout the paper. We included Table 1 summarizing the clinical trials and pre-clinical work with senotherapeutics for melanoma. While there is some published information regarding senotherapeutics with melanoma, there is still much to be done. The primary intent of the review is to bring the two fields of melanoma and senescence/aging together and encourage more research in this area.

  • I struggle with Figure 1’s contribution to this article

Response: We can move this figure to the supplement, however this figure touches on two major contributors to senescence: genome instability and oncogenic induced replication stress. It also explains how these two drivers of senescence occur in UV damaged skin and in melanocytes and why BRAFV600 missense substitutions are commonly found. A sentence clarifying this point was added (line 135-137).

  • While I understand there are distinct molecular differences, there was no discussion of the relationship between senescence and quiescence. There are a number of publications outlining how targeted inhibitors leave behind a cancer stem cell enriched/quiescent-like “minimal residual disease” (Rambow et al. Cell 2018) - these underlying concepts work hand-in-hand.

Response: We included reference to the Rambow et al. paper and discuss how their results play into senescence (line 208-222). Distinguishing the difference between senescence and quiescence in vivo is difficult considering that cells that escape from senescence have a more stem phenotype and this is similar to the neural crest stem cell population described in minimal residual disease in the Rambow et al. paper. Currently there is not enough evidence or data to directly compare senescence and quiescent in MRD populations, but we make parallel comparisons in Figure 3 and throughout the paper.

  • How does intermittent dosing (Das Thakar et al Nature 2013 and others) play a role in senescence?

Response: Intermittent dosing is also a strategy suggested for senotherapeutics in relation to cancer (lines 415-419).  We included reference to the Das Thakar et al. article and mention the intermittent dosing for senotherapeutics as well.

  • On line 179, the authors mention that MEKi was combined with RAFi to reduce paradoxical activation and secondary cancers. This needs a citation, and is not the sole reason for combination treatment.  Also, there is no mention of “paradox breaking” RAFi, or pan-RAFi.

Response: This sentence regarding combination BRAFi and MEKi was clarified to induce drug resistance and secondary cancers and referenced (lines 292-296).  The pan-RAFi have not been extensively published and very few clinical applications have been reported. If you know of relevant articles to include, please advise.

  • While touched on at the end of the “Senescence in tumorignesis and melanoma” section, the role of the immune system should be expanded. How might RAFi induced immunogeneic cell death play a role in senescence (Erkes et al. Cancer Discovery 2020)? How do checkpoint inhibitors contribute to or reduce the number/viability of senescent cells?

Response: We included reference to the Erkes et al paper and provide evidence that HMGB1 is also an established marker of senescence (lines 233-239).

The checkpoint inhibitors are not as effective as BRAFi/MEKi for BRAF mutant melanoma and triple therapy has high toxicity, therefore they were not discussed in this review.

Minor points:

  • Line 49-50: Survival of less than 10% seems to be low.

Response: It was updated with a newer statistic along with other melanoma statistics (lines 33-34 and lines 50-51).

  • Line 121-123: This part is not clear – what is embryonic lethal?

Response: It has been restated with more clarification (lines 149-150).

  • Line 252: Define immunosenescence.

Response: Immunosenescence has been defined (lines 331-336).

Reviewer 4 Report

The review is well though and presented so far. However, it need to revised by adding some more critical information.

Senescence can not be complete without the role of p53 and p16. The authors should focus a section on the interplay between p53, p16 and RTK to delineate the relation between the oncogenic drivers, tumor suppressors and senescence.

Most of time, therapy resistance and tumor relapse is an underlying reason for the epigenetic modification. so the revised version should include a section on the epigenetic regulation  of senescence.

Author Response

Thank you for your feedback.  Please find our responses to your comments below.

The review is well though and presented so far. However, it need to revised by adding some more critical information.

Senescence can not be complete without the role of p53 and p16. The authors should focus a section on the interplay between p53, p16 and RTK to delineate the relation between the oncogenic drivers, tumor suppressors and senescence.

Response: We included in the introduction a section outlining the induction of senescence and roles of tumor suppressors p53, p16 and p21 (lines 81-96). We included that oncogenic induced senescence occurs through replication stress (120-121).  In addition, we discuss that RTKs contribution is in overriding senescence by driving the pro-survival pathways (lines 178-182 and Figure 2)

Most of time, therapy resistance and tumor relapse is an underlying reason for the epigenetic modification. so the revised version should include a section on the epigenetic regulation  of senescence.

Response: We expanded the information on non-genetic contributions to drug resistance (lines 206-259) and added a new figure (Figure 3) to directly address the lack of non-genetic mechanisms of resistance.

Round 2

Reviewer 1 Report

Now the review can be acceptable for publication.

Author Response

Thank you for your feedback.